# The Value of ER∝ in the Prognosis of GH- and PRL-Secreting PitNETs: Clinicopathological Correlations

**DOI:** 10.3390/ijms242216162

**Published:** 2023-11-10

**Authors:** Roxana-Ioana Dumitriu-Stan, Iulia-Florentina Burcea, Valeria Nicoleta Nastase, Raluca Amalia Ceaușu, Anda Dumitrascu, Laurentiu Catalin Cocosila, Alexandra Bastian, Sabina Zurac, Marius Raica, Catalina Poiana

**Affiliations:** 1Department of Endocrinology, ‘Carol Davila’ University of Medicine and Pharmacy, 020021 Bucharest, Romania; 2‘C. I. Parhon’ National Institute of Endocrinology, 011863 Bucharest, Romania; 3Department of Microscopic Morphology/Histology, ‘Victor Babes’ University of Medicine and Pharmacy, 300041 Timisoara, Romania; 4Angiogenesis Research Centre, ‘Victor Babes’ University of Medicine and Pharmacy, 300041 Timisoara, Romania; 5‘Bagdasar-Arseni’ Emergency Hospital, 041915 Bucharest, Romania; 6Department of Pathology, Colentina Clinical Hospital, 020125 Bucharest, Romania; 7Department of Pathology, Faculty of Dental Medicine, ‘Carol Davila’ University of Medicine and Pharmacy, 050474 Bucharest, Romania

**Keywords:** pituitary neuroendocrine tumor, PRL-secreting PitNET, GH- and PRL-secreting PitNET, estrogen receptor alpha, prognostic factors

## Abstract

Pituitary neuroendocrine tumors (PitNETs) are divided into multiple histological subtypes, which determine their clinical and biological variable behavior. Despite their benign evolution, in some cases, prolactin (PRL) and growth hormone (GH)-secreting PitNETs may have aggressive behavior. In this study, we investigated the potential predictive role of ER∝, alongside the clinicopathological classification of PitNETs (tumor diameter, tumor type, and tumor grade). A retrospective study was conducted with 32 consecutive cases of PRL- and mixed GH- and PRL-secreting PitNETs (5 patients with prolactinomas and 27 with acromegaly, among them, 7 patients with GH- and PRL- co-secretion) who underwent transsphenoidal intervention. Tumor specimens were histologically and immunohistochemical examined: anterior pituitary hormones, ki-67 labeling index, CAM 5.2, and ER∝; ER∝ expression was correlated with basal PRL levels at diagnosis (rho = 0.60, *p* < 0.01) and postoperative PRL levels (rho = 0.58, *p* < 0.001). In our study, the ER∝ intensity score was lower in female patients. Postoperative maximal tumor diameter correlated with Knosp grade (*p* = 0.02); CAM 5.2 pattern (densely/sparsely granulated/mixed densely and sparsely granulated) was correlated with postoperative PRL level (*p* = 0.002), and with ki-67 (*p* < 0.001). The IGF1 level at diagnosis was correlated with the postoperative GH nadir value in the oral glucose tolerance test (OGTT) (rho = 0.52, *p* < 0.05). Also, basal PRL level at diagnosis was correlated with postoperative tumor diameter (*p* = 0.63, *p* < 0.001). At univariate logistic regression, GH nadir in OGTT test at diagnostic, IGF1, gender, and invasion were independent predictors of remission for mixed GH- and PRL-secreting Pit-NETs; ER∝ can be used as a prognostic marker and loss of ER∝ expression should be considered a sign of lower differentiation and a likely indicator of poor prognosis. A sex-related difference can be considered in the evolution and prognosis of these tumors, but further studies are needed to confirm this hypothesis.

## 1. Introduction

PitNETs are a group of tumors that arise from the pituitary gland. Based on cell lineage, lactotroph and somatotroph tumors come from pituitary-specific POU-class homeodomain transcription factors (PIT-1).

The prevalence and incidence of prolactin (PRL-) secreting PitNETs are approximately 50 per 100.000 and 3–5 new cases/100.000/year [1]. The percentage of GH-secreting PitNETs that co-secrete PRL varies across studies and depends on the diagnostic criteria. Also, the percentage of growth hormone (GH-) secreting PitNETs that co-secrete PRL varies across studies, but overall, the coexistence of hyperprolactinemia reaches approximately 40%.

The first line of treatment for PRL-secreting PitNETs is medical therapy with dopamine agonists (DAs), which usually respond to this treatment with PRL normalization and adenoma shrinkage. Yet, 10–15% of these neoplasms show resistance to these treatments. Surgery can serve as another choice, mainly for patients intolerant or resistant to dopamine agonists. Furthermore, surgery is also recommended as initial therapy for patients with pituitary apoplexy with severe clinical symptoms, acute intracranial hypertension, or patients with massive extrasellar extended adenomas with a high risk of visual impairment [2,3]. For refractory cases, radiotherapy, especially gamma knife radiosurgery, and temozolomide treatment are also considered.

The recent Italian Guideline for the Management of Prolactinomas recommends the resection of the adenoma by an expert pituitary surgeon for patients in the following cases: patients who do not exhibit rapid improvement of neuro-ophthalmologic impairment after two weeks of cabergoline treatment, or who are resistant/intolerant to cabergoline or other DA, who escape from DA effects, or patients who require treatment but are unwilling to take chronic medical therapy [4].

DA resistance is defined as the failure to normalize PRL levels and to achieve at least a 50% tumor size reduction at the maximally tolerated doses of DA [4]. In common clinical practice, the suggested maximum dose of cabergoline is around 4 mg per week [4]. At least 6 months on the highest tolerated DA dose is suggested as the minimum duration of treatment. Partial resistance to DA is defined as a decrease in the tumor size and prolactin levels without normalization, requiring a higher dose of DA to achieve a complete response. Complete resistance to DA is defined as failure to obtain normal prolactin values, a failure to reduce tumor size by 50%, and/or failure to regain fertility with maximum tolerated doses of DA [4].

Studies have noted some sex-related differences regarding PRL-secreting PitNETs: microprolactinomas were mainly observed in premenopausal women, whereas macroprolactinomas were more common among men older than 50 years [4].

Acromegaly is a rare disease, characterized by persistent excess of GH, which stimulates the synthesis and secretion of insulin-like growth factor 1 (IGF1). In the majority of patients, it is caused by sporadic GH-secreting PitNET. PRL- and GH-secreting PitNETs, mammosomatotroph, and mixed somatotroph-lactotroph adenomas have positive staining for GH and PRL, which are well documented in the literature for the pathological aspects, but the clinical features are poorly described [5].

Mammosomatotroph tumors arise from a single-cell population Pit-1 lineage that produces both GH and PRL. According to the latest World Health Organization (WHO) classification from 2022, GH/PRL co-secreting tumors include dimorphous PitNETs composed of GH- and PRL-secreting cells (mixed somatotroph-lactotroph tumors) but also monomorphous PitNETs with cells that produce both PRL and GH within the same cell.

### 1.1. Estrogen Receptors Expression and the Pituitary Gland

Estrogen receptors (ERs) are members of the steroid receptor gene superfamily, functioning as ligand-induced transcription factors. Two distinct isoforms of the estrogen receptor, estrogen receptor alpha (ER∝) and estrogen receptor beta (ERβ), have been identified. The proliferative effects of estrogen, mediated through its nuclear receptors, ER∝ and ERβ, have been involved in pituitary cell proliferation and tumorigenesis [6].

The molecular mechanisms underlying pituitary tumorigenesis are not completely understood. Several proposed mechanisms explain estrogen action and the development of lactotroph hyperplasia, hyperprolactinemia, and prolactinoma [7]. Estrogens can affect the differentiation of PRL cells, progenitor cells, and cells that are part of the microenvironment. Lower estrogen receptor alpha expression can be related to the sex differences observed in aggressive and malignant lactotroph tumors that are resistant to dopamine agonists [8].

Pituitary tumor transforming gene (PTTG), Myc, aldehyde dehydrogenase 1A1 (ALDH1A1), dopamine D2 receptor (D2R), mitogen-activated protein kinases (MAPK), vascular endothelial growth factor (VEGF), fibroblast growth factor 2 (FGF2), transforming growth factor β (TGFβ), and other growth factors and cytokines, such as interleukines are involved in PRL secretion, cell growth, and proliferation [7]. Also, these factors can be the basis of sex-related differences in the evolution of PRL-secreting PitNETs [8,9]. Pituitary hyperplasia, lactotroph replication, and PTTG are induced by estrogens.

Other important roles of estrogens in the pituitary include angiogenesis and regulation of adenohypophysial hormone synthesis and secretion [10]. G-protein-coupled estrogen receptor 1 (GPER)-mediated estrogen signaling seems to be more involved in antiproliferative and apoptotic actions in the pituitary but may contribute to rapid secretion of PRL under physiological conditions. GPER expression is under ER∝-mediated nuclear signaling in PRL-secreting PitNETs. In GH-secreting PitNETs, estrogen plays a secretagogue role in GH secretion that negatively modulates somatostatin receptor (SSTR) expression and reduces the somatostatinergic tone, which results in enhanced GH secretion. Also, estrogens regulate GH—IGF1 axis activity in several ways, affecting both pituitary GH secretion and peripheral hepatic IGF1 production.

### 1.2. ER∝ as a Prognosis Factor

ER∝ expression was associated with elevated proliferation markers, high tumor grade, tumor size, invasion, DA resistance, progression after multimodal therapy, and male gender [11]. Inverse correlations have been established between ER∝ expression and markers of proliferation in PRL-secreting tumors. Loss of ER∝ expression should be considered a sign of lower differentiation and an indicator of poor prognosis [11]. A significant correlation also exists between ER∝ mRNA levels and PRL levels, tumor volume, and TGFβ1 (tumor growth factor β1) mRNA levels [12,13,14].

This study aimed to evaluate the prognosis value of ER∝ in PRL- and GH-secreting PitNETs and to identify other clinicopathological correlations that can help clinicians apply personalized therapy. Few data are available in the literature regarding ER∝ and functional or non-functional PitNETs.

## 2. Results

### 2.1. Patient Characteristics

We included 32 patients with a confirmed diagnosis of PRL-secreting PitNETs and PRL- and GH-secreting PitNETs (with a female predominance, 5 patients were diagnosed with PRL-secreting PitNETs and 27 with GH- and PRL-secreting PitNETs). The mean age at diagnosis was 45.9 ± 12.8 years old.

For the patients with GH- and PRL-co-secretion, the biochemical diagnosis of acromegaly was based on the Endocrine Society Clinical Practice Guidelines [15,16,17]. All patients underwent transsphenoidal surgical intervention.

Based on radiological assessment, microadenomas were diagnosed in 16 cases, macroadenomas in 11 cases, and giant adenomas in 5 cases. Two patients had pituitary apoplexy. The mean maximal tumor diameter at diagnosis was 23.9 ± 14.5 mm.

Descriptive parameters and clinical characteristics at presentation and postoperative evaluation (3 to 6 months) are shown in Table 1.

### 2.2. Histopathological Examination

Based on the histopathological examination, 16 tumors (50%) were acidophils, 11 had a mixed pattern (34.3%), and 5 cases (15.6%) were cromophobe (Table 2). The main architectural pattern was papillary (18.7%) (Figure 1).

### 2.3. Immunohistochemistry Evaluation

We grouped the cases based on the 2022 WHO Classification (Table 3). Specimens stained for GH in 17 cases (53.1%), with intense positivity (+3) in 15 cases (46.8%), and moderate positivity (+2) in 2 cases. Tissues from 24 patients showed positivity for PRL (intense staining, +3, in 7 cases).

For the somatotroph tumors that showed dominant co-immunoreactivity higher than 10% for PRL, we considered the tumor mixed, somatotroph-lactotroph (9 cases, 28.1%). Other IHC staining combinations were mainly GH + PRL + TSH + FSH/LH in 3 cases (9.3%) or GH + PRL + FSH/LH in 2 cases (6.2%).

The Ki-67 labeling index had a median value of 3.1 ± 0.5, the majority of cases showed <3% (81.2%), and only one case had a ki-67 of 6%. In the whole group, 6 cases had a ki-67 >3%.

Based on CAM 5.2 expression, cases were divided into sparsely and densely granulated, and the sparsely granulated were the most common subtype. For the PRL-secreting PitNETs, all cases were densely granulated (Figure 2).

### 2.4. Sex-Related Differences

The study included 23 (68.8%) females and 9 male patients (28.2%). Men had larger tumors at diagnosis, with a mean maximal tumor diameter of 34 ± 13.2 mm, u9 versus female patients, with a mean diameter of 19.9 ± 13.3 mm (*p* = 0.006).

Based on the radiological evaluation, 7 female patients (21.8%) had microadenomas and male patients were diagnosed with macroadenomas or giant adenomas. (Table 4).

The majority of patients have never been treated before surgery; only 3 male patients and 1 female patient received medical treatment with carbergoline before surgery. The indication for surgery was established in the cases of PRL-secreting PitNETs resistant to medical treatment (1 female and 1 male), or in the cases with tumor apoplexy (2 male patients). Postoperatively, the surgical cure was achieved in 4 (17.3%) female patients and 1 (11.1%) male patient, with no statistical differences. Recurrence appeared in 4 female patients and 2 male patients (*p* = 0.66).

The patient’s sex was correlated with the cure rate, with an OR = 8.3 (95% CI: 1.39–49.87), and with female patients having less chance of being cured.

### 2.5. Factors Correlated with ER∝ Expression

ER∝ expression was correlated with the control of the disease under medical treatment with an OR = 0.13 (95% CI: 0.02–0.76). There were no differences between the ER∝ IR scores among female or male patients on semi-quantitative evaluation (*p* = 0.32). Data on ER∝ expression in the study population is summarized in Table 5.

Factors correlated with the ER∝ expression were the PRL level at diagnosis (rho = 0.60, *p* < 0.01) and the postoperative PRL level (rho = 0.58, *p* < 0.001) (Table 6).

Another correlation was found between IGF1 level at diagnosis and postoperative GH nadir value in the OGTT test (rho = 0.52, *p* < 0.05). Also, basal PRL level at diagnosis was correlated with postoperative tumor diameter (*p* = 0.63, *p* < 0.001).

Postoperative maximal tumor diameter was correlated with Knosp Grade (*p* = 0.02). CAM 5.2 pattern (densely/sparsely granulated/mixed densely and sparsely granulated) was correlated with postoperative PRL level (*p* = 0.002) and with ki-67 (*p* < 0.001).

Basal serum prolactin concentrations were positively associated with postoperative tumor diameter (r = 0.63, *p* < 0.001) (Figure 3).

GH level and IGF1 were correlated with the granulation pattern (Figure 4), and IGF1 level at diagnosis was in relation with the postoperative tumor diameter (Figure 5).

### 2.6. Predictors of Remission

At univariate logistic regression, GH nadir in oral glucose tolerance test (OGTT) test at diagnostic, IGF1, gender, and invasion were predictors of remission (Table 7). The stepwise regression did not identify a prognosis model (most probably, the small number of patients included in the study).

## 3. Discussion

PRL-secreting PitNETs can have variable behaviors; the majority are benign tumors but in some cases, they can have an aggressive evolution. Some clinical, pathological, and molecular factors have been identified as prognostic factors, that can help clinicians identify and apply a personalized treatment. The expression of ER∝ is one possible prognostic factor; studies have shown that the expression is lower in women and can be correlated to aggressiveness. In addition, PRL-secreting PitNETs in men are characterized by lower ER∝ expression, which can be related to higher tumor grades, resistance to treatment, and an overall worse prognosis [18,19].

Based on the five-tiered classification, taking into account invasion and proliferation, a grade 2b PRL-secreting PitNET has a 20-fold increased risk of progression compared to a grade 1a tumor. A lower ER∝ expression may enhance proliferation and determine progression to a higher grade (grade 2b) tumor [20,21,22]. Several mechanisms of tumor progression have been proposed. A hypothetical model shows that ER∝ level influences tumor incidence and progression: a high level of ER∝ induces the development of lactotroph tumors, and a low level of ER∝ reduces incidence but promotes tumor evolution to a higher grade by inducing cell proliferation and vascularization [23,24]. Discrete and sparse alterations lead to a non-aggressive phenotype. These data highlight the impact of the ER∝ expression level on genetic instability, cell growth, and vascularization, therefore explaining the prevalence of high-grade tumors and the predisposition to treatment resistance in men compared to women.

Until now, most studies on ER∝ expression have used a manual immunostaining technique. In our study, the ER∝ intensity score (IR) was higher in men versus women (1.1 ± 1.4 versus 0.8 ± 1.3, *p* = 0.32). The percentage of tumors showing ER∝ expression (PRL- or PRL- and GH-secreting PitNETs) was 34.4% in our population. Most of the resistant cases (*n* = 21) lacked ER∝ expression. Lower expression of ER∝ was observed in postmenopausal women and in men in our study (most of the patients included were postmenopausal women).

Tumors without ER∝ expression tend to be larger and tend to relapse after surgery [25]. In our study, the IR score for ER∝ was adapted from studies in breast cancer. ER∝-positive tumors are considered well-differentiated and have a lower fraction of dividing cells [26]. The expression of ER∝ may be considered a prognostic factor and can be used with other factors like the high expression of cell cycle proteins or loss of chromosomes [27]. In the normal human pituitary, ER∝ is expressed at high levels in lactotroph and gonadotroph cells [28].

Patients with positive ER∝ expression had a higher basal PRL level (ng/dL) compared with the ones with negative ER∝ expression (3299.7 ± 470, versus 6675.8 ± 203.3, *p* = 0.006). Maximal tumor diameter at diagnosis was higher in cases with ER∝ expression versus tumors with no expression (31.3 ± 19.4 versus 20 ± 9.7, *p* = 0.03). On univariate logistic regression, ER∝ expression was not a significant predictor (*p* = 0.14).

Estrogens stimulate PRL release and can differentially affect cell proliferation and PRL secretion [28,29]. The mechanism linked to the inhibition of tumor growth by ER∝ may be related to the antiproliferative effect or the antiangiogenic action of the DA treatment [30]. Unfortunately, the apoptosis induced by DA treatment may require the presence of estrogens [31].

Invasive prolactinomas may be associated with a high Ki-67/MIB-1 labeling index, indicating increased cell proliferation, although this is not a universal finding. In our study, resistant cases had a higher ki-67 labeling index compared to the responsive ones (3.1% versus 3%).

In our study, we evaluated the expression of SSTR5, which is known as the most important receptor in the regulation of PRL secretion, unlike SSTR2 [32,33,34]. In our study, we had 32% of cases that expressed SSTR5 (intensity score range: 1–3). The majority of the cases that showed SSTR5 expression did not express ER∝. Some data from clinical trials showed that treatment with somatostatin analogs can be a solution for resistant PRL-secreting PitNETs. The explanation for this effect is due to the higher affinity of Octreotide LAR and Lanreotide LAR for SSTR5 and a lower affinity for SSTR2. This treatment can have an effect on lowering PRL levels and tumor shrinkage.

In the case of mixed PRL- and GH-co-secreting PitNETs, the data available for the prognosis value of ER∝ is limited. Estrogens play a secretagogue role in GH secretion: at first, they negatively modulate SST receptor expression, reducing the somatostatinergic tone, which enhanced GH secretion. In our study, ER∝ expression was associated with PRL basal levels. No correlations were found between ER∝ expression and preoperative GH nadir in OGTT or IGF1. We found that IHC staining for ER∝ was positive in 32.1% of patients with acromegaly (GH- and PRL-secreting PitNETs).

Estrogens can suppress IGF1 levels. The supposed mechanism associated with the interference of estrogen and selective estrogen receptor modulators (SERMs) on IGF1 generation is the blockade of ER in the hypothalamic–pituitary axis, with decreased hepatic IGF1 production [35,36,37]. Estrogens upregulate liver-specific GH and GH-binding protein expression. This is how estrogens upregulate the expressions of suppressor of cytokine signaling-2 (SOCS-2) in the liver (depending on the dose), disrupting GH-induced intracellular signaling (Janus kinase2 (JAK2) phosphorylation), and thus attenuating intracellular GH signaling, leading to IGF1 reduction.

ER∝ reached higher concentrations when it was present in pituitary tumors, especially in invasive adenomas [38]. No association was found between ER∝-positive IHC staining and the size of the adenoma in acromegaly patients.

In our study, the granulation pattern was correlated with the ki-67 labeling index (*p* < 0.001), ki-67 had higher values in DG (densely granulated) cases, with a mean of 4.2% versus 3% in SG (sparsely granulated cases).

At univariate logistic regression, GH nadir in the OGTT test at diagnostic, IGF1, gender, and invasion were independent predictors of remission for the mixed GH- and PRL-secreting PitNETs.

Until now, not many studies have proposed predictive models for GH- and PRL-secreting PitNETs. The majority of the data available in the literature show that the most reliable predictors for remission are cavernous sinus invasion, ki-67 labeling index, and tumor volume. In a retrospective study that included 501 cases of patients diagnosed with functional or non-functional PitNETs (who underwent surgical treatment), one model that predicted long-term event-free survival included cavernous sinus invasion, tumor diameter ≥ 2.9 cm, and ki-67 > 3% [39]. The other model tested identified the smaller tumors at risk and included ki-67 > 3% and cavernous sinus invasion.

The time needed for biochemical remission has been another prognostic factor in some studies. In our study, the average time needed for the patients to reach remission after surgery was 45.9 weeks and 71.9% of patients were controlled for treatment after surgery. Gross total resection was possible in 84.4% of patients, and this is a very important factor that influences remission after surgery. Knosp grade 4 and partial resection were identified as independent risk factors for tumor recurrence or progression in a large study that included patients with large or giant PitNETs [40].

The growth rate of PitNETs is another predictor of remission. The preoperative growth rate was associated with age, FGFR-4 (fibroblast-growth factor receptor 4), and p27 negativity. Residual tumor volume was associated with older age, gender, and suprasellar cavernous sinus extension [41,42,43].

## 4. Materials and Methods

This retrospective, observational study was conducted following the Declaration of Helsinki and approved by the Institutional Ethics Committee of ‘C. I. Parhon’ National Institute of Endocrinology, Bucharest, Romania (Ethics Approval no. 04/24.02.2022). It included 32 patients with a confirmed diagnosis of acromegaly or prolactinoma (PRL- and mixed PRL- and GH-secreting PitNETs) in evidence at ‘C. I. Parhon’ National Institute of Endocrinology (Pituitary and Neuroendocrine Pathology Department, Bucharest, Ro-mania), who underwent pituitary neurosurgical intervention in the Neurosurgery Clinic of ‘Bagdasar Arseni’ Emergency Clinical Hospital (Bucharest, Romania), in the Neuro-surgery Clinic of ‘Colentina’ Hospital (Bucharest, Romania), in the Neurosurgery Clinic of Brain Institute, Monza Hospital (Bucharest, Romania), or NeuroHope Clinic (Bucharest, Romania).

Using the postoperative tumor paraffin blocks, we performed morphological and immunohistochemical analyses.

The inclusion criteria were: adult patients with PRL- and PRL- and GH-secreting PitNETs that underwent transsphenoidal intervention and the exclusion criteria: patients with non-functioning or other types of functional PitNETs; patients who were not eligible for transsphenoidal intervention or patients who did not have available the postoperative paraffin-embedded blocks.

All included patients with PRL-secreting PitNETs who underwent surgical intervention (patients resistant to DA, patients with pituitary apoplexy, or patients who had severe visual disturbances).

Imaging studies were performed using computed tomography (CT) or magnetic resonance imaging (MRI). Also, the patients were evaluated for cardiovascular and metabolic comorbidities, and visual field testing was performed. The patients who did not achieve biochemical control after surgery received second-line treatments: surgical re-intervention, radiotherapy, medical therapy, or combined therapy.

The postoperative tumor blocks underwent morphological and immunohistochemical analysis at the Department of Microscopic Morphology/Histology and Angio-genesis Research Centre, ‘Victor Babes’ University of Medicine and Pharmacy (Timisoara, Romania).

Short-term outcomes were determined approximately 3 months postoperatively, as follows: biochemical remission in the absence of adjuvant medical treatment and residual tumors evaluated by magnetic resonance imaging (MRI) or computed tomography (CT). For acromegaly, remission was defined as age- and gender-appropriate insulin-like growth factor-1 (IGF1) levels and GH suppression to an oral glucose challenge below 0.4 ng/mL; in patients who did not have a GH suppression test, fasting GH < 1 ng/mL was used in conjunction with IGF1 [13]. For prolactinomas, remission was defined as normalized, gender-appropriate prolactin levels [14].

Long-term outcomes and events after surgery included biochemical recurrence, radiological tumor recurrence, radiation therapy, and reintervention. Tumor recurrence during follow-up was defined as the emergence of a tumor in the context of a prior negative 3-month MRI (without residual tumor). Radiotherapy was considered for a residual tumor in the cavernous sinuses or its progression during follow-up. Biochemical recurrence was defined as the return of the hypersecretory state in patients who achieved remission at 3 months postoperatively. For patients with acromegaly who did not achieve biochemical remission, medical treatment with somatostatin receptor ligands was used as first-line medical therapy, with reintervention and/or radiation recommended in individual cases depending on tumor accessibility as well as response and tolerance to medical therapy. Patients with prolactinomas with persistent or recurrent hyperprolactinemia postoperatively were treated with dopamine agonists, while reintervention or radiation was recommended on an individual basis in patients resistant or intolerant to medical therapy.

### 4.1. Histopathological Exam

The histopathological diagnosis was established after routine staining with hematoxylin and eosin (H&E) on 3 µm sections for each case. The quality of the specimens was verified using immunostaining with vimentin (ETU Leica, clone V9, RTU). Morphological staining was performed using a Leica Autostainer XL (Leica Biosystem Newcastle Ltd., Balliol Business Park West, Benton Lane, New Castle Upon Tyne, NE 12 EW, UK). Microscopic examination was performed with a Nikon Eclipse E 600 microscope (Nikon Corporation, Tokyo, Japan). Antibodies used were from Dako Cytomation, Agilent, Santa Clara, CA, USA and Thermo Fisher Scientific, Waltham, MA, USA.

### 4.2. Immunohistochemical Staining

After the histological evaluation of the specimens stained with Hematoxylin and Eosin, the immunohistochemical hormonal profile was evaluated. The primary antibodies used were as follows: GH (Anti-GH, DakoCytomation, polyclonal rabbit anti-human, dilution 1:400), PRL (Anti-PRL, DakoCytomation, dilution 1:300), ACTH ([adrenocorticotropic hormone], Anti-ACTH, DakoCytomation, clone C93, dilution 1:50), FSH ([follicle stimulating hormone], Anti-FSH, ThermoScientific, clone FSH03, dilution 1:500), LH ([luteinizing hormone], Anti-LH, ThermoScientific, clone LH01, dilution 1:500) and TSH ([thyroid-stimulating hormone], Anti-TSH, ThermoScientific, Mouse Monoclonal Antibody, clones: TSH01 + TSH02, dilution 1:400), ER∝ (Estrogen Receptor, Clone 6F11, RTU, Leica Biosystems, dilution 1:400), Ki-67 (anti-Ki-67, Leica Biosystem, clone MM1, RTU), Cytokeratin Cam 5.2 (clone CAM5.2, RTU, Diagnostic Byosistem), Bond Epitope Retrieval Solution 1 and 2 with pH 6 and 9 were used for unmasking (Leica Biosystems, Newcastle Ltd., Newcastle Upon Tyne NE 12 8EW, UK) and 3% hydrogen peroxide was used to block endogenous peroxidase for 5 min. The next step was to incubate with the primary antibodies for 30 min (for anterior pituitary hormones) and 20 min (for ER∝). Secondary and tertiary antibodies were applied for 8 min each. The visualization was made using the Bond Polymer Refine Detection System. Incubation with 3.3 diamino-benzidine chromogen was 10 min. The counter-staining was performed with hematoxylin and applied for 5 min. This was followed by the introduction of the sections in absolute alcohol for 5 min, their drying and clarification in benzene for the same period. Mounting was carried out automatically with the Leica CV Mount, using a permanent mounting medium type Entellan.

### 4.3. Scoring of Immunohistochemical Stains

The immunohistochemical reactions were assessed at the cellular level. The immunohistochemical expression of GH, PRL, TSH, ACTH, FSH, and LH was analyzed at the cytoplasmatic level and the expression of Ki-67 and ER∝ in the nucleus. Stains for the 6 pituitary hormones were scored in a blinding fashion. The proportion score for anterior pituitary hormones was quantified according to the following criteria: score 0 (0–10% positive cells), score 1+ (10–30% positive cells), score 2+ (30–60% positive cells), and score 3+ (>60% positive cells). The intensity scores used were from 0 to 3+ (from absent to strongly stained). A staining superior to 10% was considered positive for interpreting the results. The nuclear positive cells for Ki-67 were quantified by optical optic microscopy (magnification x20) using Image J (semiautomatic evaluation, which excluded endothelial and stromal cell nuclei). The scoring system for ER∝ was calculated as the product of the percentage of positive nuclei and the intensity of the staining and received a range of values points from 0 to 6 [15]. The scoring was adapted from breast cancer.

### 4.4. Data Analysis

The frequency of the categorical variables (sex, histological type, and adenoma size—macro- or microadenomas) was presented as a percentage. For numerical variables (age at diagnosis, diameter of the lesion), mean ± standard deviation and median were used, and a Spearman’s coefficient was used to verify correlations between numerical variables. The Mann–Whitney U non-parametric test was used to compare numeric variables between the groups. The normal distribution of continuous variables was evaluated through the Kolmogorov-Smirnov test. We performed univariate analysis for the evaluation of the relationship between the considered variables and disease outcome. We performed logistic regression. The level of significance adopted for the statistical tests was 5% (*p* < 0.05). The statistical analysis was performed using IBM SPSS statistics subscription software version 29 (International Business Machines Corp., Armonk, NY, USA).

## 5. Conclusions

In our study, ER∝ was correlated with basal PRL levels at diagnosis and postoperative PRL levels. The ER∝ intensity score was lower in female patients. Postoperative maximal tumor diameter was correlated with Knosp Grade and CAM 5.2 pattern. Also, the granulation pattern was correlated with postoperative PRL level and with the ki-67 labeling index.

The IGF1 level at diagnosis was correlated with postoperative GH nadir value in the OGTT test. Basal PRL level in PRL- and mixed PRL- and GH-secreting PitNETs was correlated with postoperative tumor diameter.

At univariate logistic regression, GH nadir in the OGTT test at diagnostic, IGF1, gender, and invasion were independent predictors of remission.

In GH- and PRL-secreting PitNETs, estrogens have a complex regulatory pattern affecting hormone secretion, gonadotroph, and lactotroph cell proliferation, as well as cell apoptosis. Based on our findings, ER∝ can be used as a prognosis marker alongside the clinicopathological PitNETs classification. Unfortunately, the main limitations of the study are the small sample size and the retrospective design. Future studies on larger populations are required to further characterize ER∝ as a novel biomarker for tumor size and invasiveness.

## Figures and Tables

**Figure 1 ijms-24-16162-f001:**
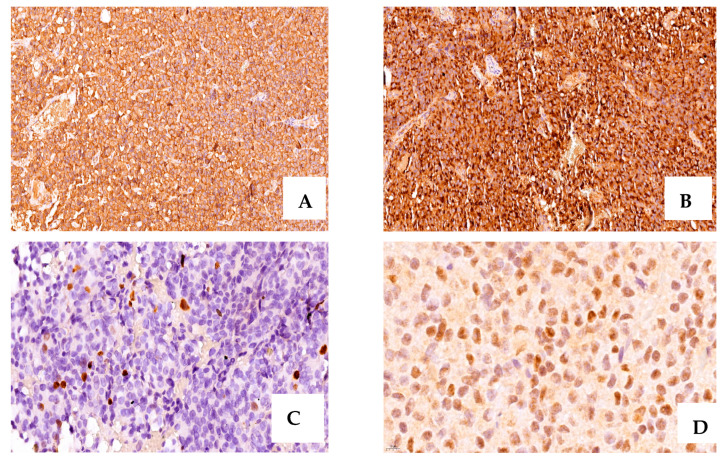
A case of acromegaly due to a GH- and PRL-secreting PitNET with positive intense IHC staining for GH (+3, ×20 magnification) (**A**), PRL (+3, ×20 magnification) (**B**), ki-67 = 4% (×40 magnification), (**C**), ER∝ positive, and moderate intensity (+2, ×80 magnification) (**D**).

**Figure 2 ijms-24-16162-f002:**
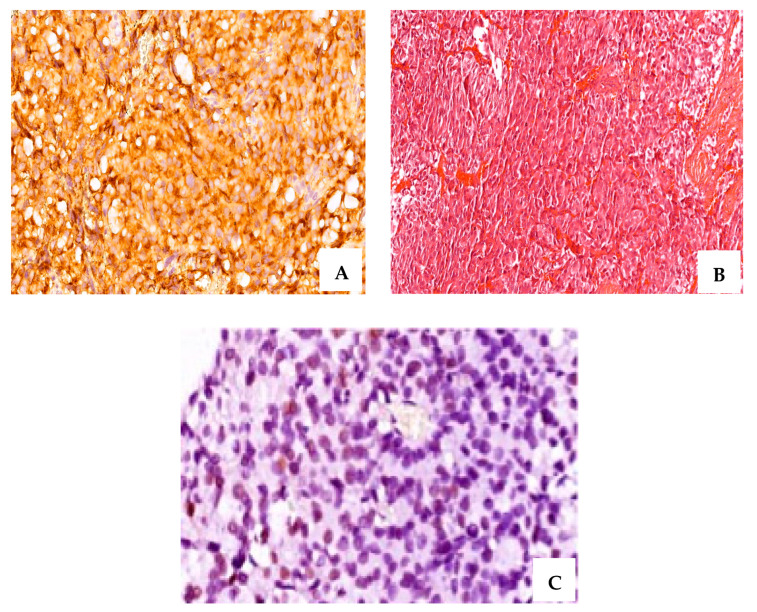
A case of resistant PRL-secreting PitNET with intense positive IHC staining for PRL (**A**), (×40 magnification); Cromophobe and acidophil tinctoriality, H&E staining (**B**) (×20 magnification); ER∝ intense positive (**C**) (+3, ×60 magnification); CAM 5.2, densely granulated pattern (**D**) (×40 magnification); ki-67 of 6% (**E**) (×20 magnification); T2-weighted magnetic resonance image sagittal plane—pituitary macroadenoma: 11/10/11 mm (postoperative) (**F**); T2-weighted magnetic resonance image coronal plane—pituitary macroadenoma (postoperative) (**G**).

**Figure 3 ijms-24-16162-f003:**
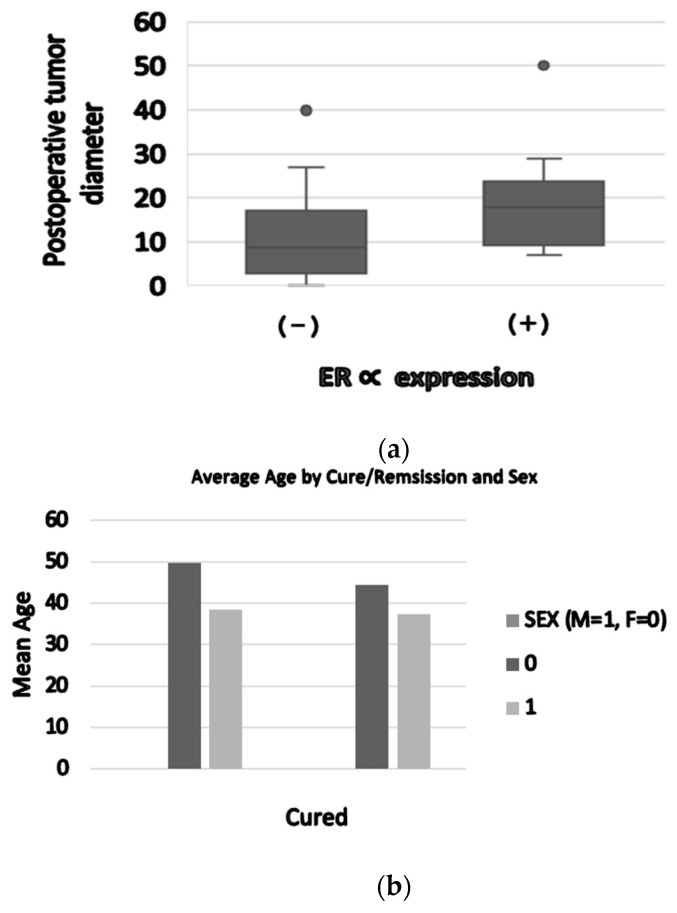
(**a**) ER∝ expression (positive/negative) was correlated with maximal postoperative tumor diameter (31.3 ± 19.4 mm versus 20 ± 9.7 mm, *p* = 0.03). (**b**) Average age by sex: female patients had a higher age compared to males (49 years old versus 37.8 years old). Patients not cured after surgery had an average age of 45.5 years old.

**Figure 4 ijms-24-16162-f004:**
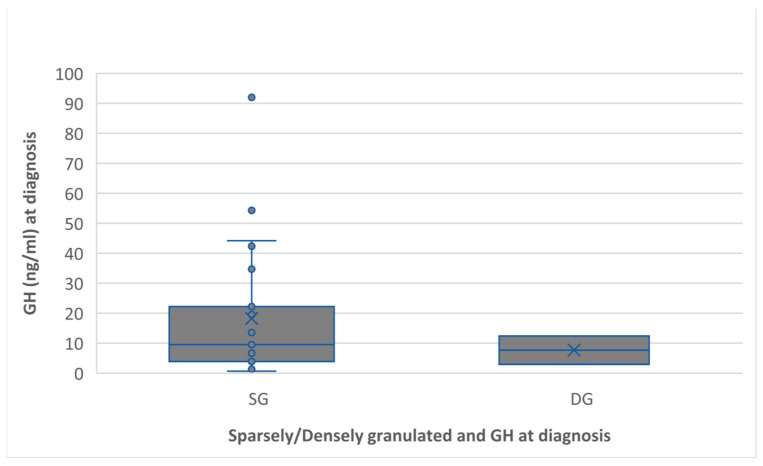
Preoperative GH level associated with the granulation pattern. The sparsely granulated pattern had a higher GH level than the densely granulated (18.2 ng/mL versus 7.6 ng/mL, *p* = 0.51).

**Figure 5 ijms-24-16162-f005:**
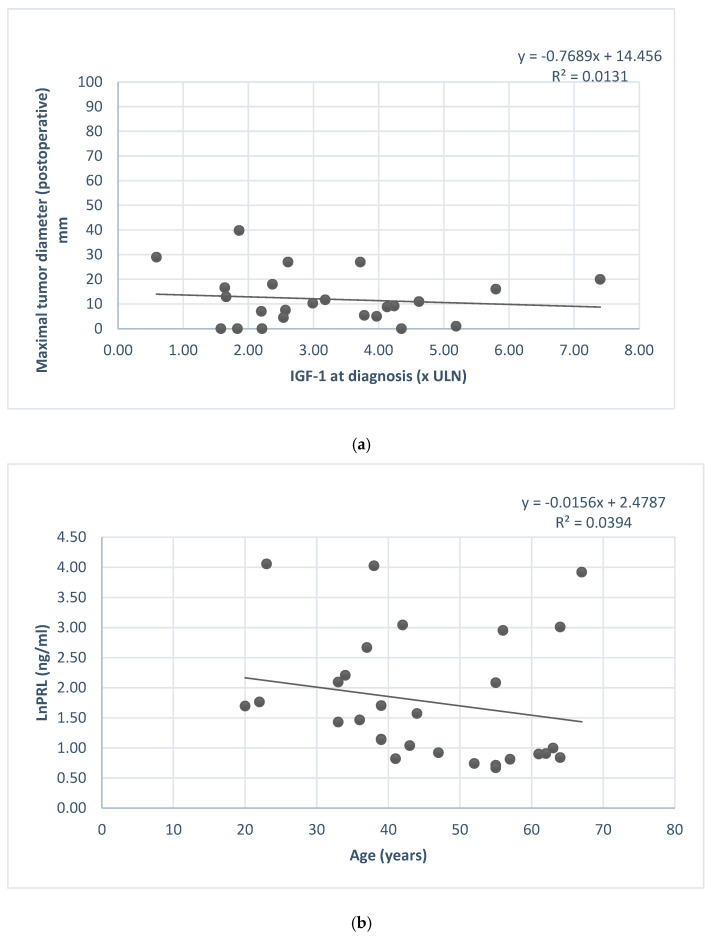
(**a**) IGF1 at diagnosis and postoperative maximal tumor diameter association. (**b**) Serum prolactin concentration (ln) and patients’ age.

**Table 1 ijms-24-16162-t001:** Descriptive parameters of the total study population.

	Distribution, n (%)
Age at diagnosis (years) *	45.9 ± 12.8
Women, n (%)	23 (68.8%)
Men, n (%)	9 (28.2%)
Biochemical diagnosis	
PRL-hypersecretion	5 (15.6%)
GH-hypersecretion	20 (62.5%)
GH- & PRL-hypersecretion	7 (21.8%)
PRL level at diagnosis, (ng/dL)	1187.5 ± 3111.6
IGF1 level at diagnosis, (×ULN)	3.1 ± 1.5
GH level at diagnosis (ng/mL)	17.4 ± 21.4
Tumor dimensions	
Maximal tumor diameter at diagnosis (mm)	23.9 ± 14.5
Microadenoma, n (%)	16 (50%)
Macroadenoma, n (%)	11 (34.3%)
Giant adenomas, n (%)	5 (15.6%)
Knosp Grade	
0, n (%)	11 (34.3%)
1, n (%)	5 (15.6%)
2, n (%)	8 (25%)
3, n (%)	5 (15.6%)
4, n (%)	3 (9.3%)
Pre-operative pituitary insufficiency, n (%)	15 (46.8%)
Apoplexy, n (%)	2 (6.2%)
Pre-surgical treatment	
Medication	5 (15.6%)
Radiotherapy	0
Follow-up	
Duration (years) *	5.4 ± 3.6
Surgical cure, n (%)	4 (12.5%)
DA resistance, n (%)	3 (9.3%)
Radiotherapy, n (%)	4 (12.5%)
Tumor progression, n (%)	4 (12.5%)
Postoperative PRL level, (ng/dL)	386.5 ± 1514.8
Postoperative IGF1 level (×ULN)	2 ± 1.3
Postoperative GH level (ng/mL)	5 ± 12
Post-surgical medical treatment, n (%)	23 (71.8%)
Post-surgical radiotherapy, n (%)	4 (12.5%)
Remission of disease, n (%)	6 (18.7%)
Control under medical treatment, n (%)	30 (93.7%)
Ki-67 labeling index	3.1± 0.5
ER∝	0.94 ± 1.3

* years and mean ± standard deviation; n = number; % = percentage; GH—growth hormone, IGF1—insulin-like growth factor 1, PRL—prolactin, ACTH—adrenocorticotropic hormone, TSH—thyroid stimulating hormone, FSH—follicle stimulating hormone, LH—luteinizing hormone, SD—standard deviation, ULN—upper limit of normal; IHC—immunohistochemistry.

**Table 2 ijms-24-16162-t002:** H&E * staining.

Categories	Distribution
Tinctoriality, n (%)	
Acidophil	16 (50%)
Cromophobe	5 (15.6%)
Mixed	11 (34.3%)
(acidophil and cromophobe)	
Pattern, n (%)	
Pseudoglandular (acinar)	2 (6.2%)
Papillary	6 (18.7%)
Trabecular	2 (6.2%)

* H&E—hematoxylin and eosin.

**Table 3 ijms-24-16162-t003:** IHC * Classification.

Adenoma Type	CAM 5.2 Expression Pattern	Pituitary Hormones	Number (%)
Somatotroph adenomas	Densely granulated	GH	2 (6.2%)
	Sparsely granulated	GH	4 (12.5%)
	Mixed pattern	GH	2 (6.2%)
Mammosomatotroph adenomas	Densely granulated	GH + PRL	1 (3.1%)
	Sparsely granulated	GH + PRL	2 (6.2%)
	Mixed pattern	GH + PRL	0
PRL− secreting adenomas	Densely granulated	PRL	2 (6.2%)
IHC			
GH		17 (53.1%)	
PRL		24 (75%)	
GH + PRL		9 (28.1%)	
GH + PRL + FSH/LH		2 (6.2%)	
GH + PRL + TSH + FSH/LH		3 (9.3%)	
GH + PRL + ACTH + LH		1 (3.1%)	

* IHC—immunohistochemistry; GH—growth hormone, PRL—prolactin, ACTH—adrenocorticotropic hormone, TSH—thyroid stimulating hormone, FSH—follicle stimulating hormone, LH—luteinizing hormone.

**Table 4 ijms-24-16162-t004:** Sex-related comparison of clinicopathological characteristics of PRL- and GH-secreting PitNETs.

	Women	Men	*p*-Value
Age (years) *	49 ± 12.3	37.8 ± 11	0.01
PRL level at diagnosis, median (ng/dL)	470.5 ± 1803	3069.5 ± 4883.6	0.02
IGF1 level at diagnosis, median (xULN)	3.1 ± 1.5	3.2 ± 1.9	0.44
Tumor maximal diameter (mm), median, at diagnosis	19.9 ± 13.3	34 ± 13.2	0.00
Invasion			
Non-invasive (%)	13 (56.5%)	4 (44.4%)	0.96
Invasive (%)	10 (43.4%)	5 (55.5%)	
Tumor dimensions			0.00
Microadenoma, n (%)	7 (21.8%)	0	
Macroadenoma, n (%)	16 (50%)	6 (18.7%)	
Giant adenoma, n (%)	2 (6.2%)	3 (9.3%)	
Ki-67 (%)	3.1 ± 0.6	3.1 ± 0.2	0.37
ER∝ (median, immunoreactive score)	0.8 ± 1.3	1.1 ± 1.4	0.32
Preoperative treatment			
Medical treatment, n (%)	1 (3.1%)	3 (9.3%)	
Radiotherapy, n (%)	0	0	
Follow-up			0.66
Surgical cure, n (%)	4 (17.3%)	1 (11.1%)	
DA resistance, n (%)	1 (11.1%)	1 (11.1%)	
Post-operative Radiotherapy, n (%)	1 (11.1%)	3 (9.3%)	
Recurrence, n (%)	4 (17.3%)	2 (6.2%)	
Post-operative tumor maximal diameter (mm), median	12.2 ± 11.5	18.3 ± 12.2	0.10

* years and mean ± standard deviation; n = number; % = percentage; GH—growth hormone, PRL—prolactin, ER ∝—estrogen alpha, DA—dopamine agonists.

**Table 5 ijms-24-16162-t005:** ER∝ expression in the study population. Univariate analysis.

	ER∝ (+)	ER∝ (-)	*p*
Women, n (%) Men, n (%)	8 (25%)	15 (46.8%)	0.23
	4 (12.5%)	5 (15.6%)	
Basal PRL level (ng/dL)	3299.7 ± 470.6	75.8 ± 203.3	0.00
Maximal tumor diameter at diagnosis (mm)	31.3 ± 19.4	20 ± 9.7	0.03
Invasiveness, n (%)	7 (21.8%)	9 (28.1%)	0.72
Gross Total Resection			0.07
- Yes, n (%)	0	5 (15.6%)	
- No, n (%)	11 (34.3%)	16 (50%)	
Cured	0	2 (6.2%)	0.29
Ki-67 (%)	3.4 ± 0.9	3 ± 0.1	0.07
SSTR 5 (+)	11 (34.3%)	21 (66.6%)	0.98

% = percentage; GH—growth hormone, PRL—prolactin, ER∝—estrogen alpha, DA—dopamine agonists, SSTR 5—somatostatin receptor 5.

**Table 6 ijms-24-16162-t006:** Correlation coefficients (Spearman’s coefficients, rho).

	Maximal Tumor Diameter at Diagnosis	Basal PRL	IGF1 at Diagnosis	Postoperative IGF1	Postoperative GH	Postoperative PRL	Maximal Postoperative Tumor Diameter	ER∝ (IR)
Basal PRL level	0.75 **	-	0.26	0.16	0.33	0.43 *	0.63 **	0.60 **
IGF1 at diagnosis	0.03	−0.23	-	0.48 *	0.52 *	0.74 **	−0.05	−0.14
Maximal tumor diameter at diagnosis	-	0.75 **	0.037	0.25	0.35	0.57 **	0.66 **	0.31
Maximal postoperative tumor diameter	0.66 **	0.63 **	−0.58	0.05	0.26	0.45 *	-	0.09
Postoperative PRL	0.57 **	0.74 **	−0.02	0.32	0.28	-	0.45 *	0.58 **
ER∝ (IR)	0.31	0.60 **	−0.14	−0.04	0.09	0.58 **	0.38 *	-

* *p* < 0.05. ** *p* < 0.001. ER∝—estrogen alpha receptor, IR-intensity score, GH—growth hormone, IGF1—insulin-like growth factor 1, PRL—prolactin, ULN—upper limit of normal.

**Table 7 ijms-24-16162-t007:** Univariate logistic regression. Predictors of remission.

Predictor		Correlation Coefficients	
	r^2^ Adjusted	beta	*p*
GH nadir in OGTT test (preoperative)	0.200	0.483	0.014
IGF1 at diagnosis	0.097	0.367	0.071
Gender	0.168	0.441	0.011
Invasion	0.135	0.404	0.022
ER∝ expression	0.040	−0.266	0.141

GH—growth hormone; PRL—prolactin; ER∝—estrogen alpha; IGF1-insulin growth factor 1; OGTT—oral glucose tolerance test.

## Data Availability

No new data were created or analyzed in this study. Data sharing is not applicable to this article.

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
