# Peer review of "The Value of ER∝ in the Prognosis of GH- and PRL-Secreting PitNETs: Clinicopathological Correlations"

_ijms, 2023, doi:10.3390/ijms242216162_

Round 1
Reviewer 1 Report
Comments and Suggestions for Authors
The paper which is well conducted, scientifically sound and proposes an interesting topic, showing how ER-alfa expression may be a useful tool to predict the natural history of PitNets. I would only suggest to a) re-elaborate Figure n.3 as the data is not very clearly understandable (maybe changing the axis or highlighting the frequencies on the columns)
Comments on the Quality of English LanguageA thorough text check for typos and English language corrections is neede
Author Response
Thank you for the comments and suggestions.
Based on the the first observation that the data in Figure n.3 is not very clearly understandable (maybe changing the axis or highlighting the frequencies on the columns) – I edited the figure and I added supplimentary explanations in the legend.
The second point: ‚A thorough text check for typos and English language corrections is needed’- we checked once again the grammar and we send the manuscript for editing to a collegue, a native speaker.
Thank you!
We hope that our revision will meet the high standards of the Journal.
Reviewer 2 Report
Comments and Suggestions for Authors
the Authors present the results of a small study investigating the predictive role of ER∝ in PitNETs. The sample size is limited, but the Authors have correctly deepened the analysis and therefore I think that this study can provide interesting results.
Unfortunately, the quality of presentation is really poor. Besides some basic mistakes (using commas to separate decimal values; reporting too many decimal values; and so on), the flow of information is really hard to follow and there are several awkward sentences. e.g., "Data available until now show that males have larger tumors, are more invasive and less sensitive to dopamine agonists treatment (DAs) and some studies confirm sex- related differences regarding PRL- secreting PitNETs". Are males more invasive?
I reccommend the Authors to carefully rework this paper, and resubmit a more readable version. there is merit in their data.
Comments on the Quality of English LanguageSee above
Author Response
Thank you for the comments and suggestions. Based on the observations: ‚Unfortunately, the quality of presentation is really poor. Besides some basic mistakes (using commas to separate decimal values; reporting too many decimal values; and so on), the flow of information is really hard to follow and there are several awkward sentences. e.g., "Data available until now show that males have larger tumors, are more invasive and less sensitive to dopamine agonists treatment (DAs) and some studies confirm sex- related differences regarding PRL- secreting PitNETs". Are males more invasive?. I reccommend the Authors to carefully rework this paper, and resubmit a more readable version. there is merit in their data.’
\We checked once again the grammar and we send the manuscript for editing to a collegue, a native speaker and reworked the parts that were hard to follow.
We hope that our revision will meet the high standards of the Journal.
Round 2
Reviewer 2 Report
Comments and Suggestions for Authors
thanks for having revised the paper according to my comments. However, the manuscript still requires improvement in terms of English. See for instance the sentence I pointed out in my first revision: "Males tend to have larger, more invasive and are resistant to dopamine agonist treatment (DAs)." It sill reads uncorrect, and this is not the only example.
Can the Authors further revise English?
Furthermore, the Authors claim that a native English colleague has revised the paper, but its identity is not disclosed in the Acknowledgments.
Comments on the Quality of English LanguageSee above
Author Response
Dear Reviewer,
Thank you so much for the comments and suggestions.
First point ‚However, the manuscript still requires improvement in terms of English. See for instance the sentence I pointed out in my first revision: "Males tend to have larger, more invasive and are resistant to dopamine agonist treatment (DAs)." It sill reads uncorrect, and this is not the only example. Can the Authors further revise English? Furthermore, the Authors claim that a native English colleague has revised the paper, but its identity is not disclosed in the Acknowledgments.’
We have revised the manuscript with the help of another colleague, and we mentioned her in the Acknowledgments. She is a pathologist and endocrinologist and has carried out her scientific activity abroad.
Second point - Methods are not adequately described - The material and methods include all steps followed for tissue processing according to international and internal protocols. These must be mentioned for the study to be validated. We revised the section and we hope now there are better explained.
Third point – Results are not clearly presented - The Results section has been modified to be more clearly understood.
The section Material and Methods, Discussion and Results were reorganized in the correct order, according to the Journal's rules.
We highlighted the parts modified with track changes.
We hope that our revision will meet the high standards of the Journal.
We hope that our submission will be of considerable interest to the readers of International Journal of Molecular Sciences.
Best regards,
Dr. Roxana Dumitriu-Stan
